# Off-axis metasurfaces for folded flat optics

Brandon Born[1], Sung-Hoon Lee[2], Jung-Hwan Song [1], Jeong Yub Lee[2], Woong Ko[2] & Mark L. Brongersma [1] ✉

The overall size of an optical system is limited by the volume of the components and the internal optical path length. To reach the limits of miniaturization, it is possible to reduce both component volume and path length by combining the concepts of metasurface flat optics and folded optics. In addition to their subwavelength component thickness, metasurfaces enable bending conventional folded geometries off axis beyond the law of reflection. However, designing metasurfaces for highly off-axis illumination with visible light in combination with a high numerical aperture is non-trivial. In this case, traditional designs with gradient metasurfaces exhibit low diffraction efficiencies and require the use of deep-subwavelength, high-index, and high-aspect-ratio semiconductor nanostructures that preclude inexpensive, large-area nanofabrication. Here, we describe a design approach that enables the use of low-index (n ≈ 1.5), low-aspect ratio structures for off-axis metagratings that can redirect and focus visible light ($\lambda$ = 532 nm) with near-unity efficiency. We show that fabricated optical elements offer a very large angle-of-view (110°) and lend themselves to scalable fabrication by nano-imprint lithography.

Metasurfaces enable the decoupling of shape and function of conventional optical components[1–5]. This opens up new possibilities for the miniaturization of optical systems through a reduction of the component count and volume[6–8]. However, the most challenging part to miniaturize is perhaps the open space between optical components. This requires a momentum-dependent transfer function that can only be achieved with a non-local metasurface[9] and there are no suggestions yet on how to achieve this in a spectrally-broadband fashion. An alternative approach is to fuse the fields of flat-optics and folded-optics[10,11] where the optical path is folded with metasurfaces, reducing the total open space volume. Furthermore, off-axis illumination of metasurfaces can greatly improve the optical contrast in imaging systems by spatially separating the zeroth order path. To achieve this, it requires metasurfaces that operate efficiently in reflection mode at large off-axis incident and diffracted angles. Unfortunately, this remains a major challenge as the diffraction efficiency diminishes severely for large angles due to angular scattering properties of the nanostructures and the limited number of such elements that can be fitted within a metagrating period[12–16]. To appreciate the challenges and identify solutions, it is of value to briefly revisit current design approaches for metasurfaces.

Metasurfaces are commonly designed by first assuming that the constituent scattering elements control the local transmission and reflection properties, independent of their neighbors. This allows for the creation of a lookup table that can map a desired phase profile[17,18]. By judiciously varying the discrete geometries of each element, one can emulate the continuous phase accumulation of curved surfaces as described by the generalized Snell's laws for reflection and refraction[19]. For metasurfaces with continuous curvature, such as metalenses[20–23], freeform reflectors[24], and retroreflectors[25,26], a larger periodic unit cell at the 2π phase-wrapping emerges. The number of elements within this supercell defines the accuracy of the phase profile and the highest performing designs naturally feature densely-spaced, high-refractive index, high-aspect-ratio, and deep-subwavelength nanostructures. Recent work has pointed out that engineering the phase gradient is not sufficient to efficiently redirect light and highlight the importance of achieving impedance matching by the metasurface to avoid spurious reflections[12–16,27]. These fabrication and impedance matching challenges become increasingly daunting for grazing incidence and large beam redirection angles, exactly what is needed for folded optics. A way to get around this challenge is to use either active local or passive, nonlocal field manipulations[12,28,29].

[1]Geballe Laboratory for Advanced Materials, Stanford University, Stanford, CA, USA. [2]Samsung Advanced Institute of Technology, Samsung Electronics Co. Ltd., Samsung-ro 130, Yeongtong-gu, Suwon-si, Gyeonggi-do 16678, South Korea. ✉e-mail: brongersma@stanford.edu

In this work, we build on the recently proposed design approach for high-efficiency, passive planar metagratings[12,13,30–32] to demonstrate a low-index centimeter-scale metasurface for folded optics. A metasurface is designed to mimic an off-axis curved mirror and is shown to achieve near-unity diffraction efficiencies. We further show how it is possible to use low-index ($n = 1.5$) materials and low-aspect-ratio (2:1) nanostructures, enabling the possible use of large-area pattering techniques such as nano-imprint lithography for the mass-production of folded flat optics.

## Results

### The folded metasurface optics concept

To demonstrate the use of metasurfaces in folded optics, this work aims to miniaturize the function of an off-axis curved mirror, as illustrated in Fig. 1a, b. Such an optical component could be integrated into cascaded folded optical systems[10] to further reduce the size. To appreciate the miniaturization benefits, we challenged ourselves to redirect a quasi-collimated beam (centered at the green wavelength $\lambda = 532$ nm) that is incident at a large grazing incidence angle ($\theta_i = 80°$) to create a focus with a large diagonal angle-of-view of 110°, corresponding to a numerical aperture (NA) of 0.82. Such large angles-of-view are critical for a number of emerging applications, including compact imaging systems and near-eye-displays. The need for a large angle-of-view thus directly translates into the need for a broad range of diffracted angles across the metasurface. Figure 1b illustrates that by using a metasurface, the optical path's cross-sectional area can be reduced by approximately 40× when compared with a conventional off-axis parabolic mirror at the same working distance. To design such a metasurface, we start from basic principles and leverage the grating equation $\sin\theta_i - \sin\theta_m = m\lambda/P$ in reflection mode. It links the incident angle $\theta_i$ of a light beam to the diffracted angles $\theta_m$ for each order $m$ given a periodic grating with a period $P$. When considering the more

general conical diffraction case[33], it is clear that a diffractive element can redirect the first order diffracted angle into any desired direction. Thus by slowly varying the pattern and orientation of a grating across its surface it is possible to emulate the continuous phase accumulation of a curved surface. It must be noted that the grating equation does not convey how much power is redirected in the desired diffraction order and how much is lost to unwanted orders. Poor efficiencies remain a critical challenge for very large redirection angles, particularly for off-resonance metagratings[32]. To achieve high diffraction efficiency, additional scatterers need to be placed to suppress unwanted orders and direct the optical power into a single desired order for a larger range of redirection angles.

In our proposed design, we place a low-refractive-index surface-relief layer on a reflective substrate, as shown in Fig. 1c. The substrate can be either fully reflective, such as a metal layer, or selectively reflective at certain frequencies or states of polarization. The latter is important for applications that require a transparent substrate, such as augmented reality eyewear. In these cases, a wire-grid polarizer or a spectrally selective narrowband Bragg mirror can be used[31]. By using large grazing incidence angles to compactify the folded optical path volume, we also enable the use of low-refractive-index materials, such as nanoparticle-free polymers compatible with nano-imprint lithography. This is a result of increased Fresnel reflectivity at grazing angles which helps to achieve the targeted interference conditions. The design aims to cancel the specular reflection (zeroth order pathway) from the mirror through destructive interference with the light reflecting from the surface relief layer. This occurs when these two pathways for the light are equal in amplitude and out of phase. Through computational optimization, we find that this can be achieved for grazing s-polarized light with low-refractive-index, low-aspect-ratio nanostructures as shown in Fig. 1c, d. With both the specular transmissive and reflective pathways suppressed, the light

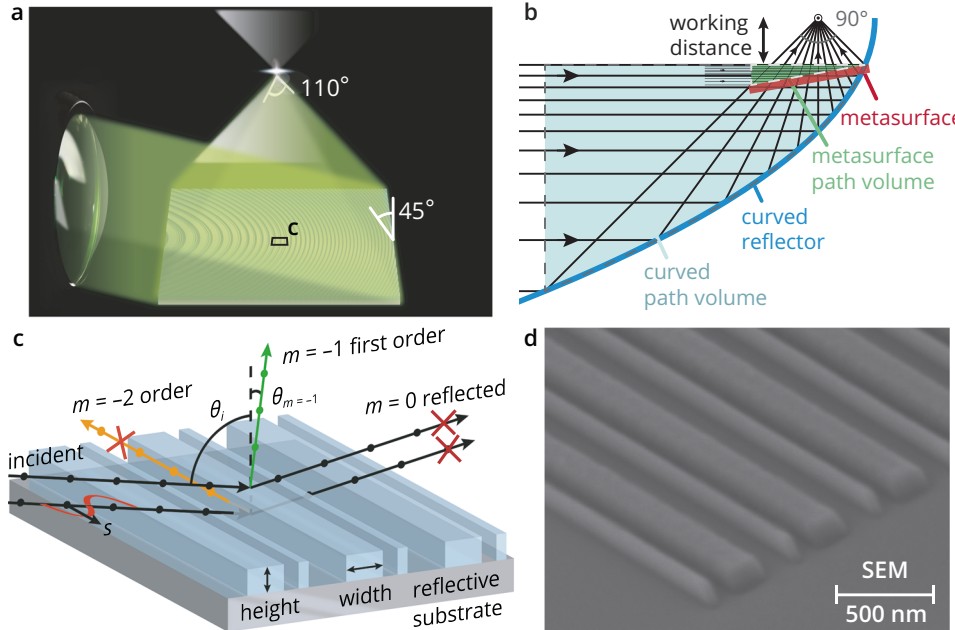

**Fig. 1 | The metasurface architecture and targeted application. a** A metasurface element capable of achieving the same function as a parabolic, metallic mirror for a very large angle-of-view of 110°, corresponding to a numerical aperture of 0.82. **b** A metasurface reduces the optical path's cross-sectional area by ~40× compared to a parabolic, metallic mirror at the same working distance. **c** A high diffraction efficiency for an off-axis beam at a grazing incidence angle of $\theta_i = 80°$, is redirected in a desired diffraction angle $\theta_{m=-1} = +5°$ by eliminating the specular reflection, $m = 0$ order, and all undesired diffraction orders. Convex optimization with the rigorous coupled wave analysis is used to optimize the number of elements per period as

well as their heights and widths to achieve the highest diffraction efficiency for s-polarized illumination. In the optimal design, all of the incident light is forced into the first-order, $m = -1$, diffracted beam by suppressing the specular reflection and other unwanted diffraction orders. **d** A scanning electron microscopy image of the metasurface reflector where there are two differently sized elements in the repeating unit cell. We will show that breaking the symmetry of the periodically repeating unit cell is critical in reaching high efficiencies for certain redirection angles.

forced into the remaining anomalously diffracted pathways. As high diffraction efficiency can be achieved in this manner, we avoid high-refractive-index resonant structures to ultimately attain a broad spectral and angular response. Next, we further optimize the geometry and spacing of each individual element within the periodic supercell to also suppress spurious diffraction and maximize the diffraction efficiency from the desired first-order pathway.

## Optimization, fabrication, and characterization of metagratings

To demonstrate the suppression of specular reflection and spurious diffraction pathways, we first consider a perfectly periodic and infinitely extended metasurface that is illuminated by narrowband light with s-polarization. Beginning with the architecture depicted in Fig. 1c, the first order, $m = -1$, diffraction efficiency is optimized. Using a congruent method to inverse design[34], each metagrating is optimized by the rigorous coupled-wave analysis (RCWA) paired with a multi-start gradient decent algorithm. Where the term metagrating is defined here as the gradient sub-section of the metasurface having the same unit cell[35]. The tunable parameters include the height, widths, spacings, and number of dielectric elements per period. The refractive index is fixed at $n = 1.57$, to represent our targeted nano-imprint compatible material, IOC-114, which is nano-particle free. The objective function is set to maximize the first-order diffraction efficiency, which will automatically minimize the other pathways. A deposited silver layer is chosen as the reflective substrate for ease of fabrication, low absorptive losses, and reduced simulation complexity. Using this optimization scheme, near-unity efficiencies can be achieved that are primarily limited by absorption in the metal.

To illustrate how we can suppress spurious diffraction, we start with a comparison of the performance of a one-element grating against a metagrating with two distinct elements per period. The one-element and two-element metagratings are both optimized at the same wavelength of $\lambda = 532$ nm, incident angle of $\theta_i = 80°$, and period of 593 nm corresponding to a desired diffraction angle of $\theta_{m=-1} = +5°$. For ease of fabrication, the grating material is an electron-beam-lithography resist, ZEP520A with a refractive index measured at $n = 1.57$ to match our targeted imprint material. In practice IOC-114 and ZEP520A may have slightly different refractive indices depending on curing conditions. The minimum feature width in the design is limited to 60 nm, corresponding to a maximum aspect ratio of 2:1 and a duty cycle of 10%. An optimal height of 120 nm is targeted, achieving high efficiencies for both the one-element and two-element metagratings. The global height of the metasurface is a tunable parameter, where the local metagrating height is fixed to the global value in accordance with the fabrication method. With our gradient decent approach, we find the optimal duty cycle to be 47% for metagratings with one element, and 39% –10% for the two-element version, given a minimum allowable 10% duty cycle for the gap between these elements. Note that both types of metasurfaces have a filling fraction near 50%.

After optimizing the metagrating, we simulate the diffraction efficiency across the visible spectral range and over a range of incident angles (50° to 90°) for the one- and two-element metagratings (Fig. 2a and inset). The one-element metagrating can effectively suppress the specular reflection and achieve a high diffraction efficiency of 72% for an 80° incident angle. However, it displays a sharp, anomalous drop in efficiency at certain combinations of incident angle and wavelength. This is attributed to be a Rayleigh-Wood anomaly, first experimentally discovered in 1902 by Wood[36]. The anomalous efficiency change was later attributed to the onset of new diffracted orders by Rayleigh[37] and to the excitation of guided surface waves by Fano[38]. Equivalently, the onset of these higher diffraction orders, with $m \leq -2$, will occur when increasing the period such that the first order's desired angle exceeds $\sin\theta_{m=-1} \geq (\sin\theta_i - 1)/2$. This is known to create a fundamental challenge for achieving high efficiency over the angular range of a large aperture diffractive lens. Fortunately, we can further increase the

degrees of freedom with this architecture by having multiple elements per period and suppress spurious diffraction. This additional degree of freedom is necessary and would improve other architectures[39] that similarly eliminate the zeroth order. The two-element metagrating shown in Fig. 2a eliminates the efficiency drop associated with the Rayleigh-Wood anomaly by suppressing the $m = -2$ diffracted order. It therefore achieves a higher efficiency of 88% at our target wavelength of operation ($\lambda = 532$ nm).

The elimination of the Rayleigh-Wood anomaly is experimentally verified in Fig. 2b. Samples of the one-element and two-element metagratings are fabricated with electron beam lithography measuring $0.5 \times 0.5$ mm in area and have the same optimized geometry as described above. The diffraction efficiencies simulated by the RCWA technique are shown as well and confirm the very different response for the one- and two-element metagratings. The simulation and experimental results match closely and indicate the superior performance of the two-element structure for near-normal exit $\theta_{m=-1} \approx 0°$. Note the data has been smoothed with a moving average to remove thermal and system noise. Some higher frequency ripples are still evident in the data which arose from periodic fluctuation in the laser source intensity over the course of the measurement. A scanning electron microscope (SEM) image of the 2-element metagrating is shown in the inset.

## Optimization, fabrication, and characterization of folded metasurface optics

Our approach to supress specular and spurious diffraction can be extended to realize high diffraction efficiencies across the full range of diffraction angles required for our high NA optical component. Figure 2c illustrates how incident light can be reflected and focused across the entire metasurface. To achieve maximal efficiencies the widths, spacings, and number of elements in each quasi-repeating period are optimized individually at each spatial location and corresponding diffraction angle to focus off-axis. The structure will ideally vary from a multi-element metagrating to a one-element metagrating moving from the left ($\theta_{m=-1} = +45°$) to right ($\theta_{m=-1} = -45°$) side of the optical component, as depicted in Fig. 2c's cross-section with a focusing angle of 90°. The bottom inset contains SEM images of fabricated samples that are optimized for diffraction angles equal to $\theta_{m=-1} = +45°, +30°, +10°,$ and $-30°$. To illustrate the necessity to vary the number of elements in the unit cell, Fig. 2d shows simulations of the diffraction efficiency versus the diffraction angle for one-element metasurface (black) up to four-element metasurface (light blue) that are individually optimized at every diffraction angle. The results show the optimal number of elements will depend on the desired diffraction angle. For example, at $\theta_{m=-1} = -30°$ an optimized one-element grating achieves better performance than an optimized three-element metagrating, given the 10% minimum duty cycle constraint. Where at $\theta_{m=-1} = +30°$, the three-element metagrating outperforms the one-element. To experimentally validate these results, eight $0.5 \times 0.5$ mm samples are optimized and measured to produce diffraction at $\theta_{m=-1} = -45°, -30°, -15°, +1°, +10°, +20°, +30°,$ and $+45°$. The diffraction efficiency is measured using $\lambda = 532$ nm, s-polarized light and the results are denoted with red dots in Fig. 2d. The experimental results are found to closely match the RCWA simulated efficiencies within the measurement error of $\pm 4\%$. The optimized dimensions of the eight experimental samples are provided in Supplementary Fig. S1 of the Supplementary Information. Also, an additional tolerance analysis is provided in Supplementary Fig. S2 of the Supplementary Information at $\theta_{m=-1} = +30°$ for each diffraction order with respect to incident angle and wavelength, illustrating suppression of higher diffraction orders with the multi-element metagrating.

To realize the off-axis-focusing metasurface reflector shown in Fig. 1a, a set of metagratings are optimized over the desired area of $2 \times 2$ cm² with a focal distance of 1 cm, resulting in a $90° \times 90°$

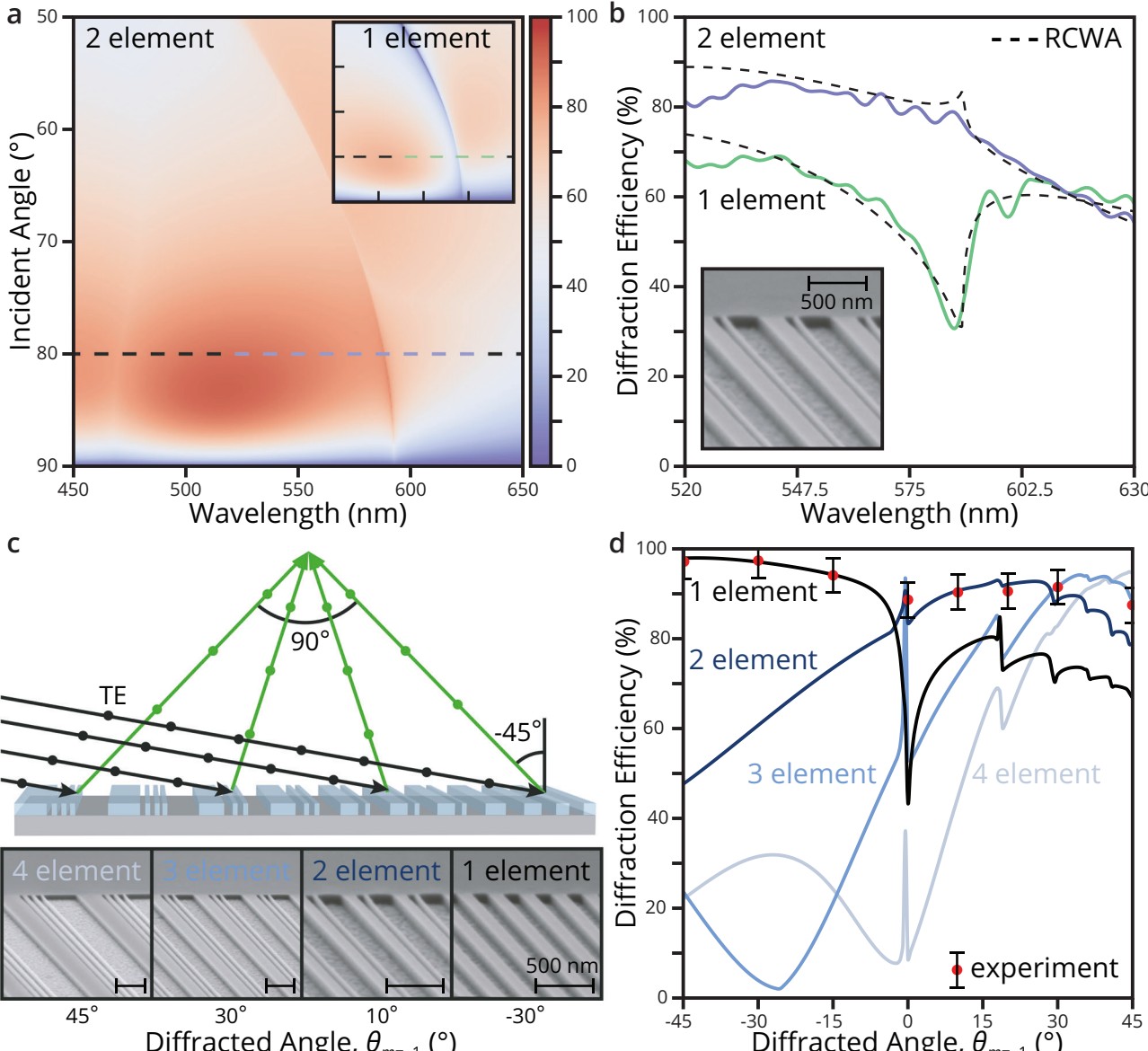

**Fig. 2 | Metagrating characterization and optimization for each diffracted angle. a** Efficiency of one-element (inset) and two-element metagratings for a broad range of wavelengths and incident angles. Both metagratings, optimized for an 80° incident angle and 5° exit angle at $\lambda = 532$ nm, succeed in suppressing specular reflection and achieve high diffraction efficiencies across a broad range of wavelengths and exit angles (88% for the optimized angle/wavelength). The two-element metagrating also suppresses higher-order spurious diffraction, avoiding efficiency losses due to the Rayleigh-Wood anomaly. The color bar is scaled from 0% to 100% diffraction efficiency for both metagratings. **b** Experimental efficiency measurements of one-element (green) and two-element (blue) metagratings in the visible range. The RCWA simulated expectations at $\theta_i = 80°$ (dashed lines) closely match the experimental results and clearly show the Rayleigh-Wood anomaly being eliminated by the two-element structure. **c** Schematic illustrating how incident s-polarized light is reflected and focused across the metasurface reflector. The optimal number of elements per quasi-repeating period varies from four to one (left to right) to enable high efficiency for any angle within the very large NA. SEM images of the relevant metagratings are in the inset below with diffracted angles equal to $\theta_{m=-1} =$ +45°, +30°, +10°, −30°. **d** Simulated efficiency versus the diffraction angle for the one-element metagrating (black) up to four-element metagrating (light blue), optimized individually. The optimal number of elements depends on the desired diffraction angle. Experimental measurements (red dots) are given with an error of ±4% for $\theta_{m=-1} =$ −45°, −30°, −15°, +1°, +10°, +20°, +30°, +45°.

angle-of-view or 110° diagonally. Up to this point, the work has investigated discrete positions along the metasurface's $y = 0$ cross-section, as seen in Fig. 2. To allow focusing in two dimensions, the surface-relief ridges are curved according to the grating equation generalized in spherical coordinates[33]. Alternatively, the position and orientation of each element can be found by simply superimposing the electric field of an off-axis plane wave with a spherical wave centered at the desired focal distance. This second method is implemented to define the metasurface supercells' pitch, position, and orientation. The metagratings are optimized by RCWA at 400 × 400 discrete points across the 2 × 2 cm surface under the assumption of an infinitely repeating

supercell period at each position. This approximation is sufficiently accurate for large centimeter-scale metasurfaces given the slowly varying phase profile of a large parabolic mirror. The structure within each supercell is determined by the nearest optimized RCWA point, effectively creating 50 × 50 μm pixels where the optimized duty cycles are constant. These duty cycles are multiplied by the local pitch such that elements still vary in width across these rounded pixels. Figure 3a illustrates the resultant metasurface, though it is a scaled down 100 × 100 μm version to make the elements visible. This rescaling is achieved by analytically recalculating the grating pattern for a 50 μm focal distance and re-interpolating to the RCWA data. Upon inspection

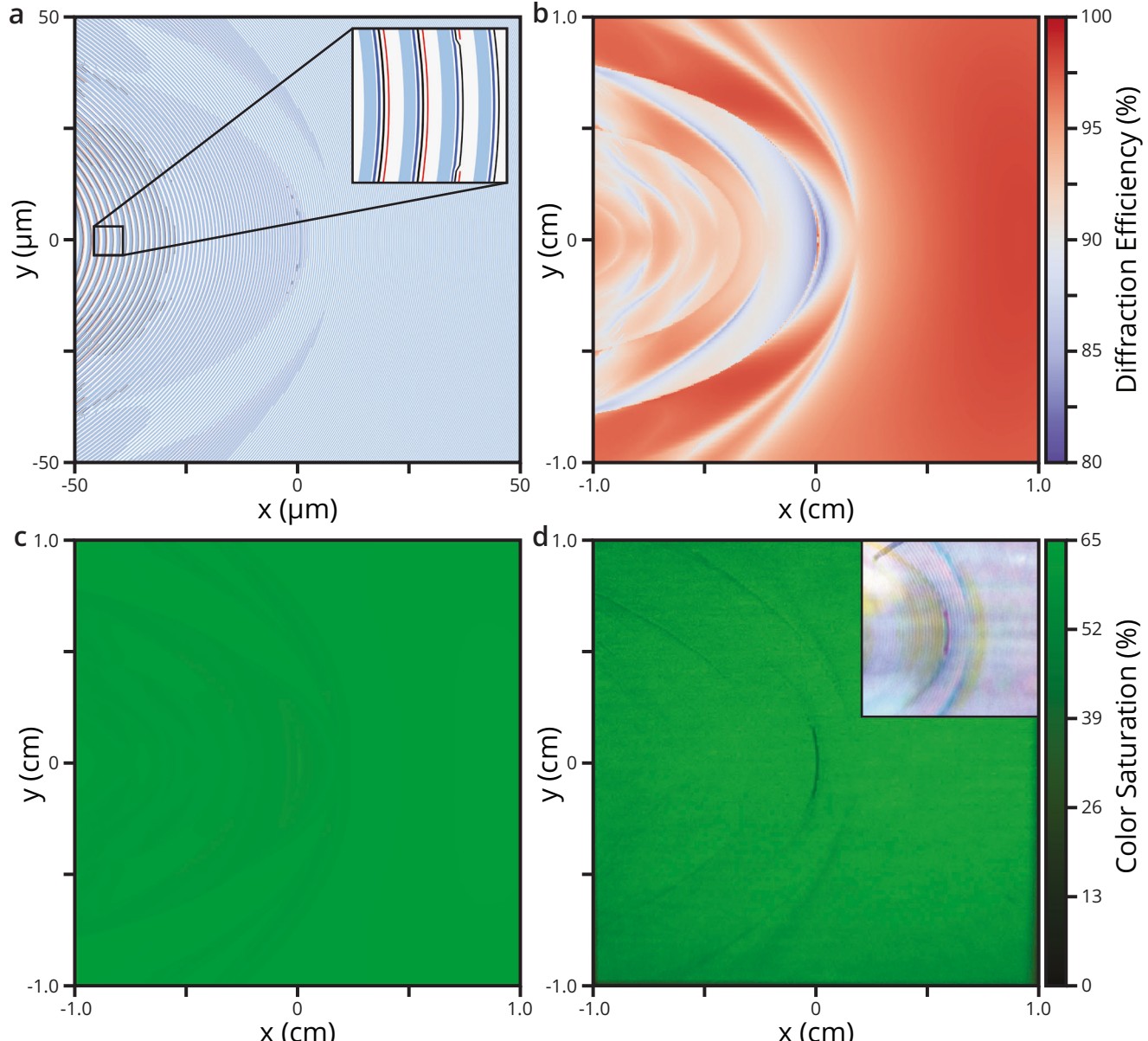

**Fig. 3 | Metasurface design and optical performance. a** Optimized metasurface, scaled at 100 × 100 μm to visualize the variation in the pitch and unit cell. Each element (blue, dark blue, black, red) slowly varies in width with transitions occurring wherever the number of elements in a 2π phase-wrapped period changes, as shown in the inset. **b** Simulated first-order diffraction efficiency of the metasurface as a function of position. The estimated mean efficiency is 95%, with a maximum of 98% and a minimum of 83%, which demonstrates high diffraction efficiencies within the targeted numerical aperture. The color bar is scaled from 80% to 100% diffracted efficiency. **c** Simulated flat-field color image derived from the diffraction efficiency. Panels **c** and **d** share the same sRGB color scale, with the maximum color saturation scaled to 65%. **d** Measured 90° × 90° flat-field color image from the fabricated 2 × 2 cm metasurface. This flat-field image represents the uniformity seen by the human eye and matches the simulation in panel c within 3.4% NRMSE. The color saturation is scaled to 65%. The inset is a multiplexed metasurface reflector capable of redirecting the primary colors of red (λ = 632.8 nm), green (λ = 532 nm) and blue (λ = 450 nm) and has a whitish appearance. An aperture is placed at the focal point to prevent cross-talk between the color channels.

of Fig. 3a, the results of optimization are seen to produce an ensemble of all the metagratings building blocks previously explored in Fig. 2. Each element is denoted in order from light blue, dark blue, black, to red. The width of each element slowly varies with transitions occurring when the number of elements in a 2π phase-wrapped period changes. This is illustrated in the inset where the number of elements decreases from four to three. Figure 3b provides the simulated first-order diffraction efficiency of the metasurface as a function of position. The estimated mean efficiency is 95%, with a maximum of 98% and a minimum of 83%. A histogram of the efficiency distribution is included in Supplementary Fig. S3 of the Supplementary Information. In Fig. 3c,

the diffraction efficiency is converted to normalized luminance in sRGB color space to illustrate the expected uniformity seen by the human eye. This is calculated using the CIE 1931 color matching function at 532 nm and converted from XYZ tristimulus values to sRGB color space values with gamma correction. These results demonstrate the design can achieve high diffraction efficiencies and uniformity for all output diffraction angles within the targeted NA.

To verify the design performance, we fabricate a 2 × 2 cm metasurface reflector by e-beam lithography, constraining ourselves to a single uniform height of 120 nm across the entire surface. With this large-scale 2 × 2 cm² sample, a flat-field calibration image is captured

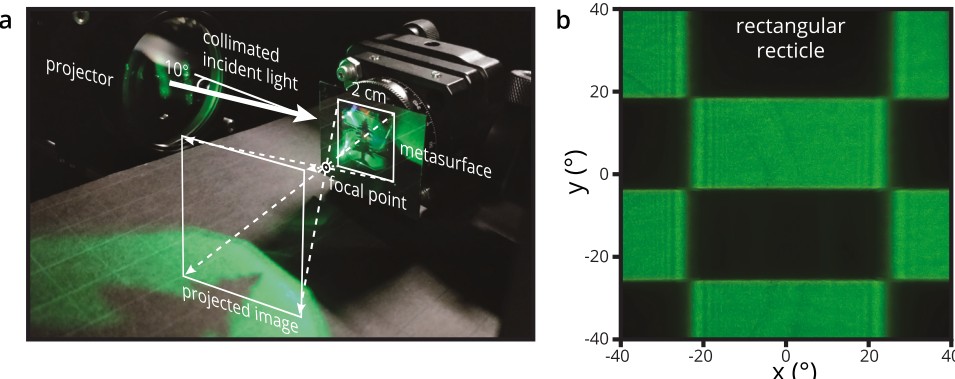

**Fig. 4 | Experimental optical setup and image characterization. a** Experimental setup and fabricated metasurface 2 × 2 cm² sample. The reticle projector is collimated 532 nm laser collimator, constructed with a fiber-coupled tunable source, aligned at 80° from normal incidence. The incident light is redirected and focused at 1 cm, with a demonstration image seen projected beyond. **b** Projected image of a checkerboard reticle used to analyze the image sharpness and quantify the MTF.

and presented in Fig. 3d. The flat-field image is measured using uniform mono-chromatic illumination and displayed in sRGB color space which is representative of the image seen by the human eye. The luminance pattern of Fig. 3d matches the sweeping arcs seen in the simulated diffraction efficiency pattern of Fig. 3b and can be directly compared to Fig. 3c. The normalized root mean square error (NRMSE) is calculated by taking the RMS of Fig. 3c subtracted by Fig. 3d, then dividing by the mean. The NRMSE is calculated to be 3.4%, indicating the fabricated samples match closely with the design given the limits of the fabrication method and the experimental setup.

**Experimental imaging with folded metasurface optics**

Figure 4a shows our experimental setup used to take the image in Fig. 3d. It employs a 532 nm laser-collimator, constructed with a fiber-coupled tunable super continuum source, which is aligned at 80° from normal-incidence to the metasurface. The laser source is expanded and collimated to provide a uniform intensity, as well as a spectrally filtered bandwidth of 5 nm. With this bandwidth and incident angle, there will be approximately 1.2 arcminutes of angular deviation due to dispersion, resulting in a low impact on sharpness. The incident light is redirected and focused 1 cm away from the metasurface, with the expanding demonstration image seen projected beyond. To capture the resulting expanded image, we place a white projector screen 5 cm away. The captured flat-field image presented in Fig. 3d corresponds to the metasurface's luminance and uniformity given a uniform input source.

We also explored the possibility of projecting 3 colors simultaneously. The inset to Fig. 3d shows a metasurface reflector that is created by multiplexing[20,40–42] metasurface reflector elements that are optimized for red ($\lambda = 632.8$ nm), green ($\lambda = 532$ nm) and blue ($\lambda = 450$ nm) wavelengths. The full color image is zoomed in to 70° × 70°. Each multiplexed curved pixel strip is 40 μm wide and lays parallel to the grating lines. The results show a similar uniform brightness, but with an approximate reduction of a factor 3 due to the multiplexing. Higher efficiencies could possibly be achieved by interleaving at the antenna level (i.e. the nanoscale)[43]. The use of such a metasurface for full-color projection would require the placement of an aperture at the focal point to prevent possible cross-talk between color channels. Another possibility would be sequential stacked layers for each color[20] and would necessitate wavelength-selective dielectric substrates for each metasurface layer.

To analyze the image sharpness, images are taken with a rectangular checkerboard reticle placed between the collimator and metasurface (Fig. 4b). The vertical and horizontal modulation transfer functions (MTF) are estimated using the edge-gradient method[44]. It quantifies how well the metasurface can transfer contrast at a

particular resolution from the projector to the image plane. From a visual inspection, it can be seen that the vertical MTF is high and has an MTF of at least 0.5 at 1 cycles per degree. However, this measurement is limited by the system MTF due to the measurement optics and mechanical constraints. The horizontal MTF is clearly lower, limited by the sinc-like intensity oscillations seen at the edge, and is measured to fall below 0.5 at approximately 0.4 cycles per degree. Measured MTF data is provided in Supplementary Fig. S5 of the Supplementary Information. The cause of the horizontal MTF degradation is attributed to the automated grating alignment algorithm used to build the layout for the e-beam patterning. Any deviation from the desired phase profile due to sparse sampling during this computation would degrade the MTF. In addition, neighboring metagratings with differing duty cycles and number of elements will impart a small phase perturbation horizontally. In the future, this could be corrected by calculating the necessary Lohmann detour phase[45,46] to cancel any difference in phase imparted to the first diffraction order. Lohmann detour phase can be applied here as a lateral translation, $0 \leq \Delta x \leq \Lambda$, of the metagrating elements within the supercell pitch, $\Lambda$, corresponding to a linear phase shift $[0, 2\pi]$ to the first diffraction order. Other source modifications could also help to provide a sharper image, such as using a scanning laser projector with a small entrance aperture stop size, i.e., a small beam diameter for a given pixel, to sample less of the global phase inaccuracy or coherent mitigation techniques to address the coherent ringing artifact[47]. Also, upon close inspection, artifacts from the 50 × 50 μm rounded pixels are visible in the flat-field image which can also degrade the image quality. A magnified image of these boundaries is included in Supplementary Fig. S4 of the Supplementary Information. Increasing the number of RCWA optimization points would help to mitigate this issue.

It has been suggested that dynamic metasurfaces can actively manipulate optical wavefronts for image projection[1–3,48]. Such methods could be integrated with the architecture presented here but would likely require significant power consumption for large-area devices. The metasurface fabricated for this work is passive, but in conjunction with a modulated scanning laser, it can emulate the optics of short-throw projectors in an ultra-compact volume. The projection of moving images is demonstrated in Supplementary Fig. S6 of the Supplementary Information.

## Discussion

The off-axis metasurface architecture introduced here uses low-aspect-ratio, low-refractive-index, dielectric elements to enable low-cost large-area manufacturing and reduced optical path volume for folded optical systems. The metagrating elements can effectively suppress specular reflection for highly off-axis incident light by tuning

a surface-relief layer placed on a reflective substrate. Higher-order spurious diffraction and Rayleigh-Wood anomalies are also suppressed by optimizing the elements with periodic supercells. These actions lead to a favorable redistribution of the power into any desired direction. To realize an off-axis focusing metasurface reflector, the surface-relief ridges are curved to allow focusing in two dimensions. The mean efficiency is 95% across the metasurface's 90°×90° angle-of-view and is experimentally verified. Ultimately, this architecture may provide an exciting new avenue for many optical applications such as beam expanders, large angle reflectors, broadband grating couplers, or could find use in folded optical systems such as AR/VR imaging devices.

## Methods

### Optical element fabrication

The metagrating elements are fabricated directly from the e-beam resist polymer ZEP520A, $n = 1.57$, which is spin coated with a uniform height of 120 nm on a silver substrate. The largest metasurface is $2 \times 2$ cm in size. The silver substrate is a polished silicon wafer, sputtered with 100 nm of silver, $n = 0.0424 + 3.10j$, and capped with 10 nm of $SiO_2$, $n = 1.465$, to prevent oxidization. This fabrication process allows for rapid prototyping while approximating the index, aspect ratio, and residual resist layer of a nano-imprinted sample for low-cost large-area manufacturing. These samples are measured with an NKT SuperK tunable supercontinuum laser source and a SpectraPro-2300i spectrometer. Given the source's 5 nm bandwidth, all simulated materials are represented with a constant refractive index measured at the reference wavelength $\lambda = 532$ nm to be consistent with the efficiency measurements.

### Optical element simulation and optimization

The metasurface elements are simulated using $S^4$, an open source RCWA solver[49]. This software is paired with a multi-start gradient decent algorithm based on MATLAB's fmincon function to optimize the highest first order diffraction efficiency. This optimization is run for discrete points along a meshed grid covering the metasurface area. The final layout interpolates over this grid to make continuous metasurface. The orientation of each metagrating is calculated by superimposing the electric field of an off-axis plane wave with a spherical wave centered at the desired focal distance. The resulting derived peaks on the surface form the grating lines, which are replaced with the optimized grating shape. The GDS file resolution is set to 1 nm, resulting in a small grating period error.

## Data availability

The data that support the plots within this paper and other finding of this study are available from the corresponding author upon request.

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

## Acknowledgements

This research was supported by the California Metaphotonics Cluster, funded by Samsung Advanced Institute of Technology (SAIT). We also would like to acknowledge funding from a Multidisciplinary University Research Initiative (FA9550-21-1-0312). B.B. was supported by the Natural Sciences and Engineering Research Council of Canada. Part of this work was performed at the Stanford Nano Shared Facilities (SNSF), supported by the National Science Foundation under award ECCS-1542152.

## Author contributions

B.B., S-H L., and M.L.B. conceived the ideas for this research project. B. B., J. Y. L., and W. K. fabricated the samples. B.B. and J-H S. performed the optical characterization. B.B., S-H L., and M.L.B. contributed to the writing of the manuscript.

## Competing interests

Some of the authors declare the following competing interest: B.B., S-H L., and M.L.B. are inventors on patent applications US16/557421 and US16/777176 that covers the use of highly off-axis metasurface grating elements for folded flat optics. The other authors declare no competing interests.
