## [Peer Review File · Nature Communications]

REVIEWER COMMENTS

Reviewer #1 (Remarks to the Author):

Review for "Off-axis Metasurfaces for Folded Flat Optics" by Born et al. (2023)

In their manuscript, Born et al. explore designing and manufacturing metagratings achieving high deflection angles and high efficiencies for visible light. The report is easy to follow, and claims are supported by the data presented in the paper. Data analysis and metrology methods are adequate and transparent. Although combining grating diffraction and local phase control has been presented to design metasurfaces before, the approach of varying the number of elements in the grating presented here gives an important boost to the efficiency while maintaining good manufacturability. With the device performance good enough for real-world applications, I feel the manuscript is of sufficient interest to the readers of a multidisciplinary journal such as Nature Communications and suggest publication after correcting some minor issues.

Minor comments and suggestions:

- 1) pg. 2: I think λ is used before it is introduced.
- 2) pg. 4: at first, the authors claim $n=1.57$ for their nano-imprint material, then they claim $n=1.57$ for ZEP520A. Is the refractive index of both materials really coincidentally the same or only similar?
- 3) pg. 4: a reference to Fig. 2a got scrambled.
- 4) pg. 4 "This is known to create a fundamental challenge toward achieving a high efficiency over the angular range of a large aperture lens.": I would say this is a challenge for a "large aperture diffractive lens".
- 5) Fig. 2a: The caption could include the exit angle that was optimized for and that the diffraction efficiency into the first diffraction order is plotted.
- 6) Fig. 2a: The color bar should have its own label.
- 7) Fig. 2b: Was the data smoothed or Fourier-filtered? If so, it should be mentioned.
- 8) Fig. 2b: Either figure or caption could include the incident angle.
- 9) pg. 6 "For this perpendicular cross-section with a 90° focusing angle, the number elements will ideally vary from ... ": I feel an "of" is missing.
- 10) pg. 6 "The correct position and orientation of each element is found by fitting the optimized geometries to the phase profile generated by superimposing the electric field an off-axis plane wave with a spherical wave centered at the desired focal distance.": a) isn't the orientation already fixed by the surface relief ridges mentioned before? If not, what does this mean exactly? b) is the position tuned to get the right detour phase or does it also change something else? As I understand it, the angle was already optimized before. This section could be expanded for ease of reading.
- 11) Fig. 3b) The color bar has no label.
- 12) Fig. 3b) The color scale could be scaled from minimum to maximum diffraction efficiency to show more detail.
- 13) Fig. 3c) The picture could use the same color scale as Fig. 2b to facilitate comparison.
- 14) Fig. 3c, inset: I feel at the present size and with all three colors overlapping, it is hard to make out details. Maybe the size could be increased, or the three different color channels could be plotted independently.
- 15) pg. 8: a reference to Fig 4b got scrambled.

16) pg. 8 and Fig. 4b: Why is the effect of the automatic grating alignment so much worse in the x than the y direction?

17) Ref. 1 has been published. Ref. 27 misses the journal name.

Reviewer #2 (Remarks to the Author):

The authors study and experimentally demonstrate metagratings for focusing a collimated beam incident from a large oblique incident angle.

The study represents some advance in metagratings to my knowledge, and the results are appealing. Yet, unless I misunderstand the meaning of the term, the primary motivation of the authors (folded metasurface optics with high efficiency metagratings) is not actually demonstrated or explored; instead, a singlet metagrating for high-NA and oblique projection is shown. The device's performance appears impressive, but the quantitative analysis is lacking here, and the jump from previous reports does not seem particularly significant to me. Given previous work and the limited scope of the demonstrations here (which do not seem thorough), the work appears too incremental and too premature to warrant publication in Nature Communications.

Primary comments:

1. The primary novelty that the authors mean to claim is not altogether clear. The authors must clarify exactly what is new about this metagrating system. For instance, high efficiency in the visible using the necessarily lower contrast systems available has been done. Both retroreflection (or $m=-1$ order) and other functionalities (e.g., $m=+1$) have been done. Spatially varying metagratings have been done. The error minimization method used is a common one. Is the novelty a combination of these? If so, what makes the authors demonstration challenging rather than a simple merging of previous capabilities? Other than the metagrating references the authors include, the authors should also look into: [10.1126/sciadv.abk3381](https://doi.org/10.1126/sciadv.abk3381) and [10.1021/acsphotonics.8b01795](https://doi.org/10.1021/acsphotonics.8b01795). They could discuss the class of devices known as “resonant domain diffractive optics”, which to my knowledge are capable of similar efficiencies (but only for narrow bandwidths) and have been around for several decades (e.g., [10.1364/OL.43.002384](https://doi.org/10.1364/OL.43.002384) for a recent work).

2. The meaning of several figures is not sufficiently explained, or are confusing, leaving their value uncertain.

2a. For instance, the axes in Fig. 3(c) need further explanation (simply calling it a “flat-field” image is not enough), especially when intended to be directly compared to Fig. 3b. My interpretation is that it is a direct color image of the metasurface when illuminated by green light (532nm with a bandwidth of 5nm), and then cropped tightly. Yet, without a colorbar, or another way to interpret this as data, the interpretation of uniformity is purely visual. More information on this characterization seems necessary to me past the mean max and min. For instance, a supplementary figure with a colorbar from scaled from the min (83%) to max (98%); histograms of the reflectivity values, etc.

2b. Fig 4(a) appears to be picture of the setup using the metagrating as an element in the illumination optics of a transparency. Presumably the transparency is removed to take the data for Fig. 3c. But none of this is clear upon first reading, and the reason for the Stanford logo in Fig 4(a) is not clear.

2c. Fig 4b seems that it could be valuable, and is claimed to be used to calculate the MTF. Yet, only two values (one for horizontal and one for vertical) are reported. This is simply not an MTF, which instead captures the observable contrast as a function of spatial frequency. It is a function. This is even clearly shown in Ref [46] that the authors cite. Again, this leaves Fig. 4b as a purely visual/qualitative indication that the device is working, but its technical and quantitative value is not satisfactory.

Secondary comments:

3. Ref. [11] does not seem to really be a folded optics reference, but instead is a metasurface singlet for the purposes of spectrometry. Perhaps the authors mean something different or broader with the term "folded optics" than systems like Ref. [10]? This should be clarified.

4. But there are other folded metasurface optics references to be added, at least this one:
[10.1021/acsphotonics.9b00744](https://doi.org/10.1021/acsphotonics.9b00744)

5. "Raleigh-Wood" should be "Rayleigh-Wood"

6. "The flat-field image of the fabricated metasurface is shown in Figure 3c and accurately matches the simulated performance in Figure 4b." The authors presumably mean "Figure 3b" not 4b.

7. The authors mention that discussion of moving images is discussed in the supplementary information, yet it is not in the Methods section.

8. Fig 1c: the graphic says "reflected", but this is imprecise. It should say "specular" or "0th order". I also believe, since it is counter to the incident direction, by convention the two diffractive orders in Fig. 1c should be -1 and -2, not +1 and +2.

9. "0a and inset" instead of "Figure 2a and inset", on page 4 (first sentence of the second full paragraph).

10. Similarly, "Also, upon close inspection of 0(b)...."

Reviewer #3 (Remarks to the Author):

Born et al. demonstrate in their manuscript an off-axis metasurface that can reflect light for large angles of incidence with high efficiency. The concept relies on a spatial variation of the grating along the surface, which was optimized by numerical methods. One advantage is the use of low-index polymers that are placed as gratings on a conductive flat substrate, which simplifies the fabrication on a larger scale. Although these metasurfaces might not become handy for mixed reality applications I can imagine applications for highly integrated optical systems where beams need to be guided or folded within restricted space. Hence, I believe that the concept is interesting and worth publication in Nature Communications.

However, I feel the text requires some major revision. Most information about the details of the structure is missing or only very generally described. A reproduction of the results is therefore not possible without doing all the numerical simulations again. I recommend the authors provide supplementary material with more detailed information about the final design of the structures and how the experimental results are obtained exactly (what was the pattern size for the homogenous pattern and how was the efficiency measured). Without giving such details the work should not be published in any scientific journal.

Here are some more specific comments that should be addressed:

- The diffraction efficiency for the 1-element grating has lower efficiency than the 2-element grating (Fig 2a). However, it depends on the diffraction angle. Here, I would recommend adding an angle to the figure caption because this is important information. It would be also helpful to show the same plots for a diffraction angle of -30 deg or less for example in the supplementary material.
- The authors talk about the visibility of the stitching error by the EBL in the flat field image 0b. Despite the wrong labeling, I am not sure where one can see the stitching. Maybe one should mark it in the image.
- The final design of the entire grating is only roughly explained and no further details are given. For me, it is not clear if each element was separately optimized based on the perfect periodicity or a few grating lines for each diffraction angle. In this context, it is also not clear how many elements were used for each of the measured points in Fig 2d. I think one should provide at least for these measurements the geometrical design parameters for the gratings in the supplementary material.
- In Fig 3a the grating lines vary in width for the curved design. Here, I recommend showing this in a better way by enlarging the inset or only using the inset and placing the entire metasurface image into the supplementary material. How is the grating size varied? Was this done based on an analytical or numerical approach?

Comments from Reviewer #1

General Comments: *In their manuscript, Born et al. explore designing and manufacturing metagratings achieving high deflection angles and high efficiencies for visible light. The report is easy to follow, and claims are supported by the data presented in the paper. Data analysis and metrology methods are adequate and transparent. Although combining grating diffraction and local phase control has been presented to design metasurfaces before, the approach of varying the number of elements in the grating presented here gives an important boost to the efficiency while maintaining good manufacturability. With the device performance good enough for real-world applications, I feel the manuscript is of sufficient interest to the readers of a multidisciplinary journal such as Nature Communications and suggest publication after correcting some minor issues.*

Reply: The authors of the manuscript are pleased to hear that the Reviewer finds our work to be of sufficient interest and deserves publication. We greatly appreciate the reviewer's attention to detail and helpful insight. We welcome this opportunity to address the corrections proposed by the Reviewer—point-by-point by way of the following replies and the corresponding revisions within the manuscript.

Comment 1: *pg. 2: I think lambda is used before it is introduced.*

Reply: The line has been corrected to “(centered at green wavelength $\lambda = 532$ nm)” to define λ and provide color context for 532 nm.

Comment 2: *pg. 4: at first, the authors claim $n=1.57$ for their nano-imprint material, then they claim $n=1.57$ for ZEP520A. Is the refractive index of both materials really coincidentally the same or only similar?*

Reply: The two materials are indeed very similar. Our team identified the nano-imprint Inkron resin IOC-114 as a preferred material candidate. Which has a $n=1.57\pm 0.02$ depending on curing conditions and wavelength. This is very close to ZEP520A, with a measured index at $n=1.57$, which we chose as our e-beam resist for this reason. The manuscript now provides more details on this topic.

Comment 3: *pg. 4: a reference to Fig. 2a got scrambled.*

Reply: All hyperlinked figure titles have been replaced to fix any scrambling.

Comment 4: *pg. 4 “This is known to create a fundamental challenge toward achieving a high efficiency over the angular range of a large aperture lens.”: I would say this is a challenge for a “large aperture diffractive lens”.*

Reply: The authors agree, and the manuscript has been updated to include the word diffractive.

Comment 5: *Fig. 2a: The caption could include the exit angle that was optimized for and that the diffraction efficiency into the first diffraction order is plotted.*

Reply: The manuscript's Figure 2a caption now includes the exit angle and wavelength it was optimized for as well as the diffraction efficiency for the optimized condition.

Comment 6: *Fig. 2a: The color bar should have its own label.*

Reply: The color bar has been explicitly labelled in the figure caption, denoting both the main figure and the inset share the same scale.

Comment 7: *Fig. 2b: Was the data smoothed or Fourier-filtered? If so, it should be mentioned.*

Reply: Yes, the data in Figure 2b was smoothed with a moving average. The manuscript has been updated to assert this.

Comment 8: *Fig. 2b: Either figure or caption could include the incident angle.*

Reply: The caption has been updated to assert the incident angle of 80° .

Comment 9: *pg. 6 “For this perpendicular cross-section with a 90° focusing angle, the number elements will ideally vary from ... “: I feel an “of” is missing.*

Reply: The sentence has now been reworded to the following, “The structure will ideally vary from a multi-element grating to one-element grating moving from the left ($\theta_{m=1} = +45^\circ$) to right ($\theta_{m=1} = -45^\circ$) side of the optical component, as depicted in Figure 2c’s cross-section with a focusing angle of 90° .”

Comment 10: *pg. 6 “The correct position and orientation of each element is found by fitting the optimized geometries to the phase profile generated by superimposing the electric field an off-axis plane wave with a spherical wave centered at the desired focal distance.”: a) isn’t the orientation already fixed by the surface relief ridges mentioned before? If not, what does this mean exactly? b) is the position tuned to get the right detour phase or does it also change something else? As I understand it, the angle was already optimized before. This section could be expanded for ease of reading.*

Reply: (a) This section of the manuscript details the generation of the curved gratings for the full two-dimensional metasurface area. Earlier in the manuscript, discrete points along a one-dimensional cross-section were investigated. The pitch and curvature of the two-dimensional gratings can be determined by solving using the grating equation, generalized to spherical coordinates, as mentioned earlier in the manuscript. Alternatively, the position and orientation of each element can be found by simply superimposing the electric field an off-axis plane wave with a spherical wave centered at the desired focal distance. In practice, this second method was implemented to define the metasurface supercell’s pitch, position, and orientation. The manuscript has been reworded for improved readability. (b) No, the grating position was not tuned to add Lohmann detour phase. This is what the authors were alluding to in their explanation of the automatic grating alignment inaccuracy of the last paragraph. This section has now been expanded and clarifies how to improve the design in future work.

Comment 11: *Fig. 3b) The color bar has no label.*

Reply: A color bar label has been added to Figure 3b.

Comment 12: *Fig. 3b) The color scale could be scaled from minimum to maximum diffraction efficiency to show more detail.*

Reply: Figure 3b has been rescaled as suggested to 80 – 100% to show more detail.

Comment 13: *Fig. 3c) The picture could use the same color scale as Fig. 2b to facilitate comparison.*

Reply: Note the Figure 3b is a measurement of diffraction efficiency or optical intensity, where original manuscript's Figure 3c is the sRGB luminance captured by our experimental setup's camera and represents what the human eye would perceive. To further facilitate comparison as suggested by the reviewer, the simulated diffraction efficiency has been converted to sRGB for a direct comparison, now labelled as Figure 3c. This image can be directly compared to Figure 3d, and normalized root mean square error (NRMSE) between the two images is calculated to be 3.4%. Further detailed information on this calculation is included in the manuscript.

Comment 14: *Fig. 3c, inset: I feel at the present size and with all three colors overlapping, it is hard to make out details. Maybe the size could be increased, or the three different color channels could be plotted independently.*

Reply: The authors have increased the size of Figure 3 and the now Figure 3d inset. It is challenging to increase the size any further given the constraints.

Comment 15: *pg. 8: a reference to Fig 4b got scrambled.*

Reply: All hyperlinked figure titles have been replaced to fix any scrambling.

Comment 16: *pg. 8 and Fig. 4b: Why is the effect of the automatic grating alignment so much worse in the x than the y direction?*

Reply: The algorithm precisely calculates the pitch, position, and orientation of each supercell position and optimizes the metagrating using the rigorous coupled wave analysis (RCWA) with a sparse grid and interpolates to the nearest grid point during the GDS file construction. As the efficiency of each metagrating is maximized, the uniformity of the electric-field amplitude is high. However, as the metagratings vary in duty cycle and number of elements, they will impart a small phase perturbation between neighboring supercells in the x direction. In the future, this could be corrected for by minimizing the phase difference between each supercell. A phase difference of zero across the metasurface could be achieved for the first diffraction order by calculating the required Lohmann detour phase to align any phase mismatch due to duty-cycle variation. Lohmann detour phase is defined as a lateral translation of the grating elements between $-\text{pitch}/2$ and $+\text{pitch}/2$ corresponding to a phase shift range of $-\pi$ to $+\pi$ respectively. This more detailed discussion has been added to the manuscript.

Comment 17: *Ref. 1 has been published. Ref. 27 misses the journal name.*

Reply: The missing journal name has been added for Lavigne et. al.

Comments from Reviewer #2

General Comments: *The authors study and experimentally demonstrate metagratings for focusing a collimated beam incident from a large oblique incident angle.*

The study represents some advance in metagratings to my knowledge, and the results are appealing. Yet, unless I misunderstand the meaning of the term, the primary motivation of the authors (folded metasurface optics with high efficiency metagratings) is not actually demonstrated or explored; instead, a singlet metagrating for high-NA and oblique projection is shown. The device's performance appears impressive, but the quantitative analysis is lacking here, and the jump from previous reports does not seem particularly significant to me. Given previous work and the limited scope of the demonstrations here (which do not seem thorough), the work appears too incremental and too premature to warrant publication in Nature Communications.

Reply: The authors of the manuscript are pleased to hear that the Reviewer finds the results impressive and we aim to address the concerns raised. Point-by-point corrections have been included by way of the following replies and the corresponding revisions within the manuscript. The presented folded metasurface has been further contextualized as a building block for cascaded folded systems. A more quantitative analysis has also been provided by way of the manuscript revision and the added supplemental information. Also, the manuscript cites the challenge of other work to achieve large incident angles and highlights the unique operational regime available with low-refractive index materials. By optimizing within this regime, the authors designed a large-area metasurface with high performance. Such an architecture could be foundational for future low-cost mass manufactured metasurfaces leveraging nano-imprint lithography.

Comment 1: *The primary novelty that the authors mean to claim is not altogether clear. The authors must clarify exactly what is new about this metagrating system. For instance, high efficiency in the visible using the necessarily lower contrast systems available has been done. Both retroreflection (or $m=-1$ order) and other functionalities (e.g., $m=+1$) have been done. Spatially varying metagratings have been done. The error minimization method used is a common one. Is the novelty a combination of these? If so, what makes the authors demonstration challenging rather than a simple merging of previous capabilities? Other than the metagrating references the authors include, the authors should also look into: [10.1126/sciadv.abk3381](https://doi.org/10.1126/sciadv.abk3381) and [10.1021/acsphotonics.8b01795](https://doi.org/10.1021/acsphotonics.8b01795). They could discuss the class of devices known as “resonant domain diffractive optics”, which to my knowledge are capable of similar efficiencies (but only for narrow bandwidths) and have been around for several decades (e.g., [10.1364/OL.43.002384](https://doi.org/10.1364/OL.43.002384) for a recent work).*

Reply: Indeed, part of the novelty is in the nontrivial combination of various insights and achievements to realize a high performance metasurface with mass-production fabrication methods. The authors have noted in the manuscript the unique challenge achieving high diffraction efficiencies at large grazing angles near the manuscript's target of 80° (something that the work by [10.1021/acsphotonics.8b01795](https://doi.org/10.1021/acsphotonics.8b01795) does not achieve). Our work also avoids high-refractive-index resonant structures to ultimately attain a broad spectral and angular response. The work of [10.1126/sciadv.abk3381](https://doi.org/10.1126/sciadv.abk3381) is interesting, specifically their use of a Bragg layer substrate. Our work however extends beyond a simple anomalous reflector to create a large-area focusing metasurfaces capable of operating in the visible spectrum, at 532 nm and for three simultaneous colors with a very large 110° field of view. The metasurface was fabricated at the large scale of traditional optics, and experimental results showed good uniformity and modulation transfer function. We believe that such a demonstration is extremely important as it shows how this type of metasurface can be applied at scale in realistic applications where optical beams need to be folded within a compact space. The authors have added the reviewer's valuable references to better place our work in context to previous works in this area.

Comment 2a: *For instance, the axes in Fig. 3(c) need further explanation (simply calling it a “flat-field” image is not enough), especially when intended to be directly compared to Fig. 3b. My interpretation is that it is a direct color image of the metasurface when illuminated by green light (532nm with a bandwidth of 5nm), and then cropped tightly. Yet, without a colorbar, or another way to interpret this as data, the interpretation of uniformity is purely visual. More information on this characterization seems necessary to me past the mean max and min. For instance, a supplementary figure with a colorbar from scaled from the min (83%) to max (98%); histograms of the reflectivity values, etc.*

Reply: The reviewer makes a good point and an explicit definition of the flat-field calibration image has been added to manuscript. The goal of Figure 3c in the manuscript is to accurately illustrate the luminance uniformity seen by the human eye. The flat-field image is captured using uniform mono-chromatic illumination and displayed in sRGB color space for this purpose. These flat-field images are commonly used to digitally correct any nonuniformity at the input panel. As this image cannot be directly compared to Figure 3b due to units, we have converted Figure 3b’s simulated efficiency results to sRGB for direct comparison. This converted image is now included in Figure 3. In addition, Figure 3b has been rescaled to better illustrate the color deviation.

Comment 2b: *Fig 4(a) appears to be picture of the setup using the metagrating as an element in the illumination optics of a transparency. Presumably the transparency is removed to take the data for Fig. 3c. But none of this is clear upon first reading, and the reason for the Stanford logo in Fig 4(a) is not clear.*

Reply: As the metasurface is acting as an off-axis focusing mirror, we are able to redirect any image or video from the projector. In the experimental setup, the authors decided to reflect a demonstration image rather than a flat-field image to illustrate the full potential of the metasurface. This demonstration image was chosen to be the Stanford logo as silhouette images are easier to capture. The manuscript now provides additional clarity about when a demonstration image is projected versus when using a flat-field image.

Comment 2c: *Fig 4b seems that it could be valuable, and is claimed to be used to calculate the MTF. Yet, only two values (one for horizontal and one for vertical) are reported. This is simply not an MTF, which instead captures the observable contrast as a function of spatial frequency. It is a function. This is even clearly shown in Ref [46] that the authors cite. Again, this leaves Fig. 4b as a purely visual/qualitative indication that the device is working, but its technical and quantitative value is not satisfactory.*

Reply: The edge gradient measurement data and calculated MTF has now been included in the Supplementary Information to corroborate the manuscript discussion. The system MTF, which sets the upper measurable limit, has also been quantified with calibration measurements and plotted.

Comment 3: *Ref. [11] does not seem to really be a folded optics reference, but instead is a metasurface singlet for the purposes of spectrometry. Perhaps the authors mean something different or broader with the term “folded optics” than systems like Ref. [10]? This should be clarified.*

Reply: The authors have replaced Reference 11 to better define the term folded optics. The folded system in Ref. 10 effectively cascades off-axis, reflective, focusing metasurfaces similar to what is demonstrated in the manuscript. However, Ref. 10’s metasurfaces are fundamentally limited in incident angle and could be further reduced in size with highly off-axis operation. And for specific applications such as in AR & VR, the highly off-axis input is a primary requirement for optical combiners. The manuscript now differentiates these cascaded folded optical systems and the folded metasurface components.

Comment 4: *But there are other folded metasurface optics references to be added, at least this one: 10.1021/acsp Photonics.9b00744*

Reply: The reference provided has been added to the manuscript, in addition to others.

Comment 5: *“Raleigh-Wood” should be “Rayleigh-Wood”*

Reply: The manuscript has been updated with the correct spelling as indicated.

Comment 6: *“The flat-field image of the fabricated metasurface is shown in Figure 3c and accurately matches the simulated performance in Figure 4b.” The authors presumably mean “Figure 3b” not 4b.*

Reply: The figure reference has been corrected to Figure 3b.

Comment 7: *The authors mention that discussion of moving images is discussed in the supplementary information, yet it is not in the Methods section.*

Reply: A supplementary section has been submitted with the manuscript with the moving image demonstration.

Comment 8: *Fig 1c: the graphic says “reflected”, but this is imprecise. It should say “specular” or “0th order”. I also believe, since it is counter to the incident direction, by convention the two diffractive orders in Fig. 1c should be -1 and -2, not +1 and +2.*

Reply: The figure has been updated to state “ $m = 0$ order”. The sign convention of Figure 1c and the manuscript text has been changed to express m as negative numbers. Where “first order” is used in the text to denote the $m = -1$ order.

Comment 9: *“0a and inset” instead of “Figure 2a and inset”, on page 4 (first sentence of the second full paragraph).*

Reply: All hyperlinked figure titles have been replaced with text to prevent this issue.

Comment 10: *Similarly, “Also, upon close inspection of 0(b)....”*

Reply: All hyperlinked figure titles have been replaced with text to prevent this issue.

Comments from Reviewer #3

General Comments: *Born et al. demonstrate in their manuscript an off-axis metasurface that can reflect light for large angles of incidence with high efficiency. The concept relies on a spatial variation of the grating along the surface, which was optimized by numerical methods. One advantage is the use of low-index polymers that are placed as gratings on a conductive flat substrate, which simplifies the fabrication on a larger scale. Although these metasurfaces might not become handy for mixed reality applications I can imagine applications for highly integrated optical systems where beams need to be guided or folded within restricted space. Hence, I believe that the concept is interesting and worth publication in Nature Communications.*

However, I feel the text requires some major revision. Most information about the details of the structure is missing or only very generally described. A reproduction of the results is therefore not possible without doing all the numerical simulations again. I recommend the authors provide supplementary material with more detailed information about the final design of the structures and how the experimental results are obtained exactly (what was the pattern size for the homogenous pattern and how was the efficiency measured). Without giving such details the work should not be published in any scientific journal.

Reply: The authors of the manuscript are pleased to hear that the Reviewer finds our work interesting and deserves publication. We welcome this opportunity to address the corrections proposed by the Reviewer in a point-by-point fashion by way of the following replies and the corresponding revisions within the manuscript. We have included a large addition of supplementary information in our submission with more details on the final design of the structures and how the experimental results were obtained.

Comment 1: *The diffraction efficiency for the 1-element grating has lower efficiency than the 2-element grating (Fig 2a). However, it depends on the diffraction angle. Here, I would recommend adding an angle to the figure caption because this is important information. It would be also helpful to show the same plots for a diffraction angle of -30 deg or less for example in the supplementary material.*

Reply: The diffracted angle and wavelength (for which the grating was optimized at) has been added to the Figure 2a caption. The region of focus where Rayleigh-Wood anomalies occur is mainly for positive angles based on our sign convention. A plot of the diffracted efficiency for each diffraction order over wavelength and incident angle has been added to the supplemental material for the metagrating optimized to +30° exit angle at $\lambda = 532$ nm.

Comment 2: *The authors talk about the visibility of the stitching error by the EBL in the flat field image 0b. Despite the wrong labeling, I am not sure where one can see the stitching. Maybe one should mark it in the image.*

Reply: All hyperlinked figure titles have been replaced to fix this mislabeling. A separate zoomed-in image has been included in the supplemental material to highlight the pixelation discussed in the manuscript. Also, a correction was made with respect to the size and ascribed root cause.

Comment 3: *The final design of the entire grating is only roughly explained and no further details are given. For me, it is not clear if each element was separately optimized based on the perfect periodicity or a few grating lines for each diffraction angle. In this context, it is also not clear how many elements were used for each of the measured points in Fig 2d. I think one should provide at least for these measurements the geometrical design parameters for the gratings in the supplementary material.*

Reply: The authors have revised the manuscript to clarify how the final design was achieved. In summary, the metagratings are optimized by rigorous coupled wave analysis (RCWA) at 400×400 discrete points across the surface under the assumption of an infinitely repeating supercell period at each position. The structure within each supercell is determined by the nearest optimized RCWA location, where the optimized duty cycles are multiplied by the local pitch. Upon inspection of Figure 3a, the results of the optimization are seen to produce an ensemble of all the metagrating building blocks previously explored in Figure 2. To further aid the reader in visualizing these building blocks that make up the final design, the geometrical design parameters for each of the experimentally measured gratings in Figure 2d have been added to the Supplementary Information as requested.

Comment 4: *In Fig 3a the grating lines vary in width for the curved design. Here, I recommend showing this in a better way by enlarging the inset or only using the inset and placing the entire metasurface image into the supplementary material. How is the grating size varied? Was this done based on an analytical or numerical approach?*

Reply: In an effort to help readers see Figure 3a, the number of figures per column has been reduced from 3 to 2 to increase the figure size. For the paper's publication the authors will ensure to submit high-resolution images, such that readers are able to digitally zoom in and inspect the entire area with high fidelity. The metasurface size is varied from 2×2 cm to 100×100 μm for Figure 3a by analytically recalculating the grating line orientations to ensure focusing at 50 μm , and numerically re-interpolating to the RCWA data. The manuscript has been updated to further elaborate how this was achieved.

REVIEWERS' COMMENTS

Reviewer #1 (Remarks to the Author):

Second Review for "Off-axis Metasurfaces for Folded Flat Optics" by Born et al. (2023)

The authors have addressed my points and improved the manuscript. If the other reviewers agree, I believe the manuscript can be published in Nature Communications after fixing the following points:

1) the used diffraction order is sometimes quoted as $m=1$ and "first-order" and other times as $m=-1$. In the response to reviewer 2, comment 8, the authors claim "first-order" means $m=-1$, which is a piece of information the reader does not have and is somewhat cumbersome. This should be carefully harmonized everywhere in the paper, together with the signs in the equation in line 74.

2) in response to my comment 13 in review 1, the authors converted Fig. 3c to sRGB (see also lines 237-242). While this allows comparison (why, e.g., is the semi-circle at coordinate 0, 0 bright in the simulation and dark in the data?) it hides some detail. Maybe the original Fig. 3b could be added to the SI to remedy this.

Minor comments:

3) line 38: is an "of" missing?

4) line 93: should "for" be "from"?

5) line 96: is an "is" missing?

6) lines 123-125: it seems like the height is optimized but lines 139-140 say "an optimal height of 120 nm", I either misunderstand or height should not be considered a tunable parameter.

7) lines 204-205: the ° symbol is missing.

8) line 206: should "valid" be "validify"?

9) lines 304-305: I feel using a smaller pupil projector would lead to a decrease in the bending angle of the metasurface, which is one of the main selling points of the paper, therefore, I would omit this statement.

Reviewer #2 (Remarks to the Author):

In my view, the authors have now made a sufficiently compelling case for publication, namely that the combination of challenges overcome represent a significant step towards practical application of meta-gratings. I agree that the combination of (1) being in the visible, (2) having large enough area to be useful in actual applications, (3) being made of low to moderate refractive index materials compatible with large area manufacturing, (4) having very high oblique incident angle of near 80 degrees, and (5) having a large range of deflection angles spanning 110, is nontrivial and worth reporting, even if most (and maybe all) of the individual aspects have been reported before. I believe this can be appreciated by a broad readership, and by experts in the field. Otherwise, the authors have also clarified the text and achievements, and addressed quantitative shortcomings of the previous version of the study. I recommend publication as is.

Minor comments:

1. The authors stated that they updated Fig. 1c to change "Reflected" to " $m=0$ ", but the figure still shows "Reflected". It is also not specified in the figure, though it is described in the main text.

2. There is a broken reference on line 200.

Reviewer #3 (Remarks to the Author):

The authors spent some minimal effort in improving the manuscript. At least it brings the paper to the level required for a publication. Nevertheless, I think the paper is worth publishing in Nature Communications. The raised questions are addressed that a better understanding of the samples and their design is now possible. Hence, I recommend the publication without further modifications.

Comments from Reviewer #1

Comment 1: *The authors have addressed my points and improved the manuscript. If the other reviewers agree, I believe the manuscript can be published in Nature Communications after fixing the following points:*

1) the used diffraction order is sometimes quoted as $m=1$ and “first-order” and other times as $m=-1$. In the response to reviewer 2, comment 8, the authors claim “first-order” means $m=-1$, which is a piece of information the reader does not have and is somewhat cumbersome. This should be carefully harmonized everywhere in the paper, together with the signs in the equation in line 74.

Reply: The authors thank the reviewer for all their previous feedback. Their input has greatly improved the manuscript, and we are glad to hear the reviewer recommends the work for publication in Nature Communications. To address their first comment, the authors have revised the manuscript to harmonize the link between $m=-1$ and the usage of “first order” diffraction more carefully. This is done by linking them in the Fig 1c, the figure caption, and in the manuscript. The grating equation has been corrected based on the sign conventions used and the critical angle equation for the Rayleigh-Wood anomaly has been checked.

Comment 2: *In response to my comment 13 in review 1, the authors converted Fig. 3c to sRGB (see also lines 237-242). While this allows comparison (why, e.g., is the semi-circle at coordinate 0, 0 bright in the simulation and dark in the data?) it hides some detail. Maybe the original Fig. 3b could be added to the SI to remedy this.*

Reply: The original Fig 3b is still there (as Fig 3b) but had been rescaled from originally 0-100% to now 80-100% to show more detail. This rescaling does make it harder to compare with Fig 3d directly but with Fig 3c now supplied, that figure can be used for direct compared. The authors believe this is a good compromise between providing high detail diffraction efficiency simulations (Fig 3b) and a low detail sRGB image (Fig 3c) to compare simulation and experimental results.

Comment 3: *line 38: is an “of” missing?*

Reply: An “of” has been added to this sentence.

Comment 4: *line 93: should “for” be “from”?*

Reply: The word “for” has been replaced to this sentence.

Comment 5: *line 96: is an “is” missing?*

Reply: An “is” has been added to this sentence.

Comment 6 *lines 123-125: it seems like the height is optimized but lines 139-140 say “an optimal height of 120 nm”, I either misunderstand or height should not be considered a tunable parameter.*

Reply: The global height of the entire metasurface is a tunable parameter. But local height for each individual metagrating is not a tunable parameter. The manuscript has been reworded to more clearly convey this.

Comment 7: *lines 204-205: the ° symbol is missing.*

Reply: An ° symbol has been added to this sentence.

Comment 8: *line 206: should “valid” be “validify”?*

Reply: The word “valid” has been replaced to this sentence.

Comment 9: *lines 304-305: I feel using a smaller pupil projector would lead to a decrease in the bending angle of the metasurface, which is one of the main selling points of the paper, therefore, I would omit this statement.*

Reply: The project entrance pupil size, i.e., the aperture stop diameter, does not limit the projector’s field-of-view or the bending angle range of the metasurface, but instead limits the beam diameter for a given pixel. The experimental setup uses a large coherent wavefront source combined with a reticle projector. Using a laser projector with a small beam diameter, i.e., pupil size / aperture stop, would sample less phase inaccuracy for any given pixel. Also introducing other coherent mitigation techniques in the projector could help to remove the ringing artifact seen in Fig 4b. The manuscript is revised to better clarifies this.

Comments from Reviewer #2

General Comments: *In my view, the authors have now made a sufficiently compelling case for publication, namely that the combination of challenges overcome represent a significant step towards practical application of meta-gratings. I agree that the combination of (1) being in the visible, (2) having large enough area to be useful in actual applications, (3) being made of low to moderate refractive index materials compatible with large area manufacturing, (4) having very high oblique incident angle of near 80 degrees, and (5) having a large range of deflection angles spanning 110, is nontrivial and worth reporting, even if most (and maybe all) of the individual aspects have been reported before. I believe this can be appreciated by a broad readership, and by experts in the field. Otherwise, the authors have also clarified the text and achievements, and addressed quantitative shortcomings of the previous version of the study. I recommend publication as is.*

Reply: The authors thank the reviewer for all their previous feedback. Their input has greatly improved the manuscript, and we are glad to hear the reviewer finds the work compelling in their eyes for publication in Nature Communications.

Comment 1: *The authors stated that they updated Fig. 1c to change “Reflected” to “m=0”, but the figure still shows “Reflected”. It is also not specified in the figure, though it is described in the main text.*

Reply: Fig. 1c has been relabeled with “m = 0 Reflected” and m = 0 has been added to the Figure caption.

Comment 2: *There is a broken reference on line 200.*

Reply: The link specified has been replaced, hopefully fixing it if it is an issue for other computers.

Comments from Reviewer #3

General Comments: *The authors spent some minimal effort in improving the manuscript. At least it brings the paper to the level required for a publication. Nevertheless, I think the paper is worth publishing in Nature Communications. The raised questions are addressed that a better understanding of the samples and their design is now possible. Hence, I recommend the publication without further modifications.*

Reply: The authors thank the reviewer for all their previous feedback. Their input has greatly improved the manuscript, and we are glad to hear the reviewer finds the work worthy in their eyes for publication in Nature Communications.